# Comparison of Treatment Patterns and Clinical Outcomes by Gender in Locally Advanced Head and Neck Squamous Cell Carcinoma (KCSG HN13-01)

**DOI:** 10.3390/cancers15020471

**Published:** 2023-01-12

**Authors:** Yun-Gyoo Lee, Eun Joo Kang, Bhumsuk Keam, Jin-Hyuk Choi, Jin-Soo Kim, Keon Uk Park, Kyoung Eun Lee, Keun-Wook Lee, Min Kyoung Kim, Hee Kyung Ahn, Seong Hoon Shin, Hye Ryun Kim, Sung-Bae Kim, Hyo Jung Kim, Hwan Jung Yun

**Affiliations:** 1Department of Internal Medicine, Kangbuk Samsung Hospital, Sungkyunkwan University School of Medicine, Seoul 03181, Republic of Korea; 2Department of Internal Medicine, Korea University Guro Hospital, Seoul 08308, Republic of Korea; 3Department of Internal Medicine, Seoul National University Hospital, Seoul 03080, Republic of Korea; 4Department of Hematology-Oncology, Ajou University Hospital, Suwon 16499, Republic of Korea; 5Department of Internal Medicine, SMG-SNU Boramae Medical Center, Seoul 07061, Republic of Korea; 6Department of Hemato-Oncology, Keimyung University Dongsan Medical Center, Daegu 42601, Republic of Korea; 7Department of Hematology and Oncology, Ewha Women’s University Hospital, Seoul 07985, Republic of Korea; 8Department of Internal Medicine, Seoul National University College of Medicine, Seoul National University Bundang Hospital, Seongnam 13620, Republic of Korea; 9Department of Hematology-Oncology, Yeungnam University Medical Center, Daegu 42415, Republic of Korea; 10Department of Internal Medicine, Gachon University Gil Medical Center, Incheon 21565, Republic of Korea; 11Department of Internal Medicine, Kosin University Gospel Hospital, Busan 49267, Republic of Korea; 12Department of Internal Medicine, Yonsei Cancer Center, Yonsei University College of Medicine, Seoul 03722, Republic of Korea; 13Department of Internal Medicine, Asan Medical Center, University of Ulsan College of Medicine, Seoul 05505, Republic of Korea; 14Department of Internal Medicine, Hallym University Sacred Heart Hospital, Anyang 14068, Republic of Korea; 15Department of Internal Medicine, Chungnam National University Hospital, Daejeon 35015, Republic of Korea

**Keywords:** locally advanced head and neck squamous cell carcinoma, gender, clinical outcome

## Abstract

**Simple Summary:**

Though women account for approximately 30% of newly diagnosed head and neck cancer, women have comprised only 17% of the study population in landmark clinical trials so far. Caution is therefore required when applying research outcomes directly to women in actual clinical practice. We hypothesized that there is no difference in treatment strategies and their effect on survival in treating locally advanced head and neck squamous cell carcinoma (LA-HNSCC) in the real world. We aimed to compare multidisciplinary treatment modalities and their outcomes by sex in 445 patients with stage III to IVB LA-HNSCC. In our overall and propensity-matched cohorts, there were no significant differences in the treatment strategy or OS by gender. In the present era, in which a multidisciplinary approach is emphasized, we conclude that there is no apparent sex-based disparity in the treatment modalities and outcomes in treating LA-HNSCC.

**Abstract:**

We aimed to compare treatment modalities and outcomes by gender in patients with locally advanced head and neck squamous cell carcinoma (LA-HNSCC). We characterized the sex-specific differences and compared the overall survival (OS) between male and female patients in a multicenter cohort of LA-HNSCC. To minimize the observed confounding, propensity score matching was utilized. The study included 445 patients; 385 (86.5%) were men and 60 (13.5%) were women. In terms of age, smoking habits, drinking habits, and primary tumor locations, there was a significant imbalance in sex before the matching. Propensity score matching yielded 60 patient pairs, with no statistical difference between the sexes in terms of their characteristics. As for the treatment strategies, there were no significant differences between the sexes before (*p* = 0.260) and after (*p* = 0.585) the propensity score matching. When comparing the survival probabilities between the sexes, OS was not significantly different in the overall (HR 1.02; 95% CI 0.59–1.76; *p* = 0.938) and propensity-score-matched population (HR 1.46; 95% CI 0.68–3.17; *p* = 0.331). These results suggest that there was no difference in prognosis by gender in the treatment modalities and outcomes of LA-HNSCC in real-world practice.

## 1. Introduction

In newly diagnosed head and neck cancer, women account for approximately 30% of incident cases [1,2]. However, a sex-based participation disparity has been observed in clinical trials investigating chemotherapy in head and neck squamous cell carcinoma (HNSCC). Clinical trials of HNSCC under the direction of the National Comprehensive Cancer Network (NCCN) guidelines underrepresented women, with women comprising only 17% of such patients [3]. In two recent phase III trials evaluating immune checkpoint inhibitors as second-line therapies for recurrent or metastatic HNSCC, about 17% of included patients were female [4,5]. In trials for locally advanced HNSCC (LA-HNSCC), approximately 11~13% of patients were female [6,7,8]. For studies on induction chemotherapy in LA-HNSCC, only 7~10% of included patients were female [9,10].

Despite this sex-based participation disparity, few studies have analyzed the effect of this disparity on clinical outcomes [2]. Genetic and biologic differences between the sexes are known to determine the efficacy and tolerability of chemotherapy [11,12,13]. For clinical trials of induction chemotherapy in LA-HNSCC, however, the chemotherapy protocols were developed without considering sex differences [10,14,15,16,17]. Furthermore, these studies did not report sex-specific clinical outcomes. Caution is therefore required when applying research outcomes directly to women in actual clinical practice. Until now, there has been no research on whether there are differences in treatment strategies and their effect on survival in treating LA-HNSCC.

Our study aimed to characterize the sex-specific differences of multidisciplinary treatment modalities in a large nationwide LA-HNSCC cohort. Specifically, we investigated whether the patient’s sex affects the clinical outcomes of LA-HNSCC.

## 2. Patients and Methods

### 2.1. Study Population

The Korea Cancer Study Group (KCSG) cohort consisted of 445 patients in total with clinical stage III to IVB LA-HNSCC (according to the American Joint Committee on Cancer Staging 7th edition) who were recruited between January 2005 and December 2015 at 13 referral hospitals in South Korea. All participating hospitals operate their own head and neck cancer multidisciplinary teams, consisting of head and neck surgeons, medical oncologists, radiation oncologists, pathologists, and radiologists.

The eligibility criteria for the LA-HNSCC cohort were an age older than 20 years and biopsy-proven squamous cell carcinoma of the oropharynx, oral cavity, hypopharynx, larynx, or other locations. Other sites included maxillary sinus, nasal cavity, ethmoid sinus, and unknown primary squamous carcinoma. Human papilloma virus (HPV) testing was not obligatory, and its positivity was based on the results of p16 expression by immunohistochemistry or HPV DNA using real-time polymerase chain reaction according to the guidelines of each participating hospital. Patients with nasopharyngeal cancer were excluded.

The study population was divided by sex to compare characteristics and study outcomes.

### 2.2. Study Objectives

The primary objective was to compare the baseline characteristics and LA-HNSCC treatment patterns by sex. The secondary objective was to compare progression-free survival (PFS) and overall survival (OS) by sex. PFS was defined as the time from diagnostic date of HNSCC to disease recurrence, progressive disease, or death of any cause. OS was calculated from the date of diagnosis to death, regardless of the cause.

### 2.3. Statistical Analyses

We compared the baseline categorical and continuous variables using the chi-squared or Fisher’s exact test and Student’s unpaired *t*-test, respectively. The time-to-event outcomes were estimated using the Kaplan–Meier method and compared using log-rank tests. Multivariate Cox regression was used to determine which variables were prognostic indicators. For building a multivariate Cox model, forward selection stepwise regression with a threshold *p*-value of 0.10 was used. We tested the interaction for variables that might have affected the impact of sex in the multivariate analysis. We tested the proportional hazard assumptions statistically and graphically.

To minimize the observed confounding bias in these retrospective analyses, we made adjustments using the propensity score matching method for the following demographics: age, performance status, smoking history, alcohol history, primary tumor location, HPV status, T classification, and N classification. To develop the propensity-score-matched pairs for the male and female patients, we adopted the nearest-neighbor matching algorithm without replacement, providing a 1:1 match.

After matching with the propensity score, we compared the baseline categorical variables with McNemar’s test and the continuous variables with the paired *t*-test. To compare the time-to-event outcomes by sex, we used stratified log-rank tests. All reported *p*-values were 2-sided, and *p*-values < 0.05 were considered to indicate statistical significance. All the analyses were performed using Stata 16.1 software (Stata Corp LP, College Station, TX, USA). The institutional review board approved this study in the main hospital (IRB-H-1304-089-481) and in each participating hospital.

## 3. Results

### 3.1. Patient Characteristics

Among the 445 patients of the LA-HNSCC cohort, 385 (86.5%) were men and 60 (13.5%) were women. The median age was 62 years (range, 24–88 years) for the men and 55 years (range, 27–81 years) for the women (*p* < 0.001). The proportion of patients aged 70 years or older was twice as high in the men than in the women. The proportion of former or current smokers and drinkers was significantly higher in the men than in the women (*p* < 0.001). The most common location for the primary tumor was the oropharynx for the men and women. Laryngeal cancer was more frequent in the men, and oral cavity cancer was more frequent in the women (*p* = 0.008). The performance status, tumor differentiation, and tumor and nodal classification were well balanced by sex. Table 1 summarizes the demographics by sex. Propensity score matching yielded 60 patient pairs. In this subset of 120 patients, there were no statistical differences between the men and women in terms of patient characteristics (Table 1).

### 3.2. Comparison of Treatments

According to the treatment modality, 229 (51.5%) patients underwent definitive concurrent chemoradiation therapy (CCRT), and 187 (42.0%) underwent surgery. Among patients who underwent surgery, 14.2% of patients received adjuvant CCRT and 13.7% received adjuvant radiotherapy. No patients received adjuvant chemotherapy after surgery. The remaining 29 (6.5%) patients did not receive adequate treatment with curative intent. Approximately 45.0% (103/229) and 17.1% (32/187) of those who intended to undergo definitive CCRT and surgery, respectively, underwent induction chemotherapy (IC). As for the treatment strategies, there were no significant differences between the women and men before (*p* = 0.260) or after (*p* = 0.585) the propensity score matching (Table 2). Among the patients undergoing definitive CCRT, the female patients were more likely to undergo IC than the male patients.

Table 3 shows the details of the treatment modalities. For the female and male patients, the preferred IC regimen was DP (docetaxel and cisplatin), TPF (docetaxel, cisplatin, and fluorouracil), FP (fluorouracil and cisplatin), and other therapies (*p* = 0.529). The median number of IC cycles was three for both sexes (*p* = 0.906). The best overall response to IC was also similar between the female and male patients (*p* = 0.117). Of the 305 patients who underwent CCRT, there were no significant differences in the preferred regimen, total radiation dose, or best overall response between the female and male patients (Table 3).

### 3.3. Comparison of Outcomes

The median follow-up duration for the overall population was 39.3 months (95% CI 35.4–43.1), and 113 patients died during the follow-up. PFS was not significantly different (HR 1.20; 95% CI 0.54–2.64; *p* = 0.063) between the female and male patients. A median OS was not reached. When comparing the survival probabilities by gender, OS was not significantly different (HR 1.02; 95% CI 0.59–1.76; *p* = 0.938) (Figure 1A). In the propensity-score-matched population, PFS was not significantly different (HR 0.32; 95% CI 0.51–3.41; *p* = 0.573). OS was also similar between the female and male patients (HR 1.46; 95% CI 0.68–3.17; *p* = 0.331) (Figure 1B). When comparing the survival probabilities between the CCRT and surgery groups, PFS and OS were not significantly different between the female patients (HR 0.56; 95% CI 0.13–2.52; *p* = 0.450 by PFS) (HR 0.97; 95% CI 0.31–3.05; *p* = 0.952 by OS) (Figure 2A) and the propensity-matched male patients (HR 0.32; 95% CI 0.08–1.28; *p =* 0.108 by PFS) (HR 1.24; 95% CI 0.36–4.25; *p* = 0.733 by OS) (Figure 2B).

The multivariate analyses for mortality showed that drinking and other primary sites were two significant predictors for poor survival in the female patients with LA-HNSCC (Table 4).

## 4. Discussion

This study aimed to compare treatment strategies and their outcomes between men and women in a nationwide cohort for LA-HNSCC. To accomplish this, we compared data on patients who underwent definitive-intent treatment according to multidisciplinary recommendations, after using propensity score matching to make the two sexes more similar. In our overall and propensity-matched analyses, there were no significant differences in the treatment strategy, PFS, or OS resulting from sex differences. In the present era, in which a multidisciplinary approach is emphasized [18], our study indicates no definitive proof of sex discrepancies in treatment strategies and outcomes in LA-HNSCC.

Significant differences between the proportion of women enrolled in HNSCC trials and those with HNSCC in the general population have long been noted [19,20]. Fortunately, this gap in the proportion has narrowed from 14.9% (1985–1989) to 8.4% (2015–2017) [3] because the proportion of women among new cases of HNSCC has decreased by 0.6% every 5 years while the proportion of women in US clinical trials has increased 0.3% every 5 years [3]. In lung cancer, where men have a 2–3-fold higher incidence than women, the underrepresentation of women due to enrollment disparity has also been alleviated [21,22]. Therefore, sex-based participation disparity in clinical trials can be expected to be mitigated in the future.

Differences between men and women in terms of LA-HNSCC treatment strategies have been reported. Based on US NCCN-cited HNSCC chemotherapy clinical trials, women are less likely to undergo definitive chemoradiotherapy than men as opposed to definitive radiotherapy [3]. For locally advanced oropharyngeal cancer, Sher et al. found that women were less likely to undergo any type of chemotherapy (but not IC), as per data from the National Cancer Database [23]. In our study, women were more likely to undergo IC when the definitive treatment strategy was concurrent chemoradiotherapy. However, there was no significant difference in survival outcomes between women and men before and after the propensity score matching. We therefore found that the difference in chemotherapy preference did not affect overall treatment outcomes.

Alcohol use is one of the most common risk factors for developing head and neck cancers [24,25]. In female populations with LA-HNSCC, alcohol use and other primary sites were two poor prognostic factors in our multivariable analyses (Table 4). However, alcohol use was not a prognostic factor in our previous analyses, which included 445 patients [2]. This could be interpreted as evidence that alcohol use has a particularly unfavorable effect on female patients. Unfortunately, the statistical power was not high enough to be conclusive due to the small number of female patients included (n = 60). Therefore, future studies including a large number of female patients are warranted. 

Our results highlight that sex-based treatment disparity and outcomes are not apparent in LA-HNSCC patients. However, the greater part of large-scale clinical trials targeting LA-HNSCC have not reported sex-specific trial outcomes. In the randomized, open-label phase III trials TAX323 and TAX324, which evaluated induction chemotherapy with cisplatin and fluorouracil alone or in combination with docetaxel in LA-HNSCC, the proportion of women in the study population was 10.3% in TAX323 and 16.3% in TAX324 [10,14,15]. Similarly, other studies targeting patients with recurrent or metastatic HNSCC have not reported sex-specific outcomes. In the KEYNOTE-048 study, a phase III trial evaluating the role of pembrolizumab and/or chemotherapy (platinum and 5-fluorouracil, cetuximab with chemotherapy) in untreated local, incurable recurrent, and metastatic HNSCC, women accounted for approximately 17% of all patients [26]. The CheckMate-141 study, which assessed the efficacy of nivolumab among patients with platinum-refractory recurrent HNSCC, also had a study population in which 17% were women. However, these studies did not report sex-specific clinical outcomes and did not include sex as a subgroup in the sub-analyses. There is therefore a need for continued interest in providing data on outcomes according to sex in future large-scale studies.

We need to address several limitations of our study. First, retrospective data collection cannot avoid selection bias. However, we attempted to reflect real-world practice by recruiting 445 patients from 13 nationwide referral hospitals for our study population. Second, the differences in characteristics and treatment preferences between men and women confer a confounding bias. To minimize these biases, we performed propensity score matching, adjusting for possible confounders that were likely to affect the treatment decision for the two sexes. However, our statistical matching technique could not control for unmeasured confounders. Therefore, the proportional inclusion of female patients in clinical trials is essential for generalizing the trial results to women. Lastly, we could not retrieve some important information regarding patient characteristics such as smoking, alcohol history, and HPV status. Future studies of a prospective nature will be necessary.

## 5. Conclusions

In our study, we found that there was no difference in treatment modalities by sex and their outcomes in real-world practice of LA-HNSCC management. 

## Figures and Tables

**Figure 1 cancers-15-00471-f001:**
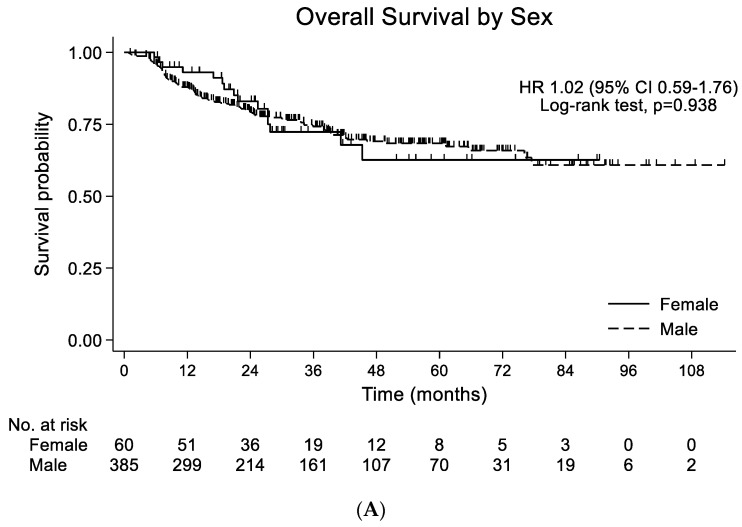
(**A**) Overall survival by sex in entire population (n = 445). (**B**) Overall survival by gender in propensity-score-matched population (n = 120).

**Figure 2 cancers-15-00471-f002:**
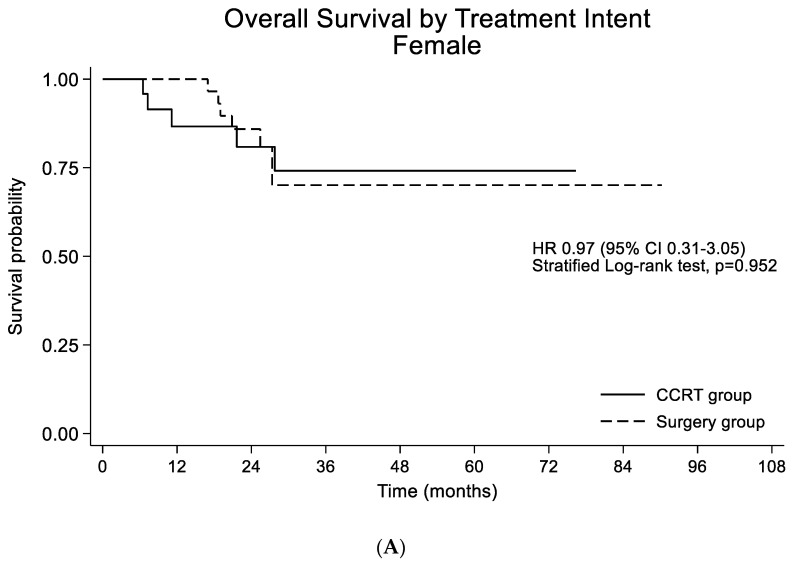
(**A**) Overall survival by treatment strategy in the female participants (n = 60). (**B**) Overall survival by treatment strategy in the propensity-score-matched male participants (n = 60).

**Table 1 cancers-15-00471-t001:** Baseline characteristics of locally advanced head and neck squamous cell carcinoma by sex.

Characteristics	Sex		*p*	Total N = 445	Propensity-Score-Matched		*p*
	Female N = 60	Male N = 385			Female N = 60	Male N = 60	
Median age, years [range]	55 [27–81]	62 [24–88]	<0.001	61 [24–89]	55 [27–81]	55 [24–74]	0.739
Elderly population			0.064				0.752
Age < 70 years	54 (90.0%)	308 (80.0%)		362 (81.4%)	54 (90.0%)	55 (91.7%)	
Age ≥ 70 years	6 (10.0%)	77 (20.0%)		83 (18.6%)	6 (10.0%)	5 (8.3%)	
ECOG PS			0.308				0.537
0	4 (6.7%)	51 (13.3%)		55 (12.4%)	4 (6.7%)	9 (15.0%)	
1	27 (45.0%)	188 (48.8%)		215 (48.3%)	27 (45.0%)	24 (40.0%)	
2	2 (3.3%)	15 (3.9%)		17 (3.8%)	2 (3.3%)	2 (3.3%)	
3	0 (0.0%)	4 (1.0%)		4 (0.9%)	0 (0.0%)	0 (0.0%)	
Unknown	27 (45.0%)	127 (33.0%)		154 (34.6%)	27 (45.0%)	25 (41.7%)	
Smoking history			<0.001				0.907
Never	41 (68.3%)	58 (15.1%)		99 (22.3%)	41 (68.3%)	40 (66.7%)	
Former	4 (6.7%)	130 (33.8%)		134 (30.1%)	4 (6.7%)	6 (10.0%)	
Current	4 (6.7%)	103 (26.8%)		107 (24.0%)	4 (6.7%)	3 (5.0%)	
Unknown	11 (18.3%)	94 (24.4%)		105 (23.6%)	11 (18.3%)	11 (18.3%)	
Alcohol history			<0.001				0.160
Does not drink	37 (61.7%)	83 (21.6%)		120 (27.0%)	37 (61.7%)	27 (45.0%)	
Does drink	8 (13.3%)	155 (40.3%)		163 (36.6%)	8 (13.3%)	14 (23.3%)	
Unknown	15 (25.0%)	147 (38.2%)		162 (36.4%)	15 (25.0%)	19 (31.7%)	
Primary location			0.008				0.125
Oropharynx	24 (40.0%)	167 (43.4%)		191 (42.9%)	24 (40.0%)	30 (50%)	
Oral cavity	22 (36.7%)	84 (21.8%)		106 (23.8%)	22 (36.7%)	17 (28.3%)	
Hypopharynx	7 (11.7%)	57 (14.8%)		64 (14.3%)	7 (11.7%)	5 (8.3%)	
Larynx	1 (1.7%)	56 (14.6%)		57 (12.8%)	1 (1.7%)	6 (10.0%)	
Other	6 (10.0%)	21 (5.5%)		27 (6.1%)	6 (10.0%)	2 (3.3%)	
Tumor differentiation			0.736				0.769
Good	12 (20.0%)	56 (14.6%)		68 (15.3%)	12 (20.0%)	11 (18.3%)	
Moderate	20 (33.3%)	139 (36.1%)		159 (35.7%)	20 (33.3%)	22 (36.7%)	
Poor	10 (16.7%)	56 (14.6%)		66 (14.8%)	10 (16.7%)	12 (20.0%)	
Not assessed	18 (30.0%)	134 (34.8%)		152 (34.1%)	18 (30.0%)	15 (25.0%)	
T classification			0.148				0.350
T1	13 (21.7%)	53 (13.8%)		66 (14.8%)	13 (21.7%)	12 (20.0%)	
T2	29 (48.3%)	143 (37.1%)		172 (38.7%)	29 (48.3%)	23 (38.3%)	
T3	8 (13.3%)	86 (22.3%)		94 (21.1%)	8 (13.3%)	7 (11.7%)	
T4a/b	10 (16.6%)	101 (26.2%)		111 (24.9%)	10 (16.6%)	18 (30.0%)	
Unknown	0 (0.0%)	2 (0.5%)		2 (0.5%)	0 (0.0%)	0 (0.0%)	
N classification			0.103				0.706
N0	8 (13.3%)	44 (11.4%)		52 (11.7%)	8 (13.3%)	6 (10.0%)	
N1	27 (45.0%)	113 (29.4%)		140 (31.5%)	27 (45.0%)	24 (40.0%)	
N2	25 (41.7%)	220 (57.1%)		245 (55.1%)	25 (41.7%)	30 (50.0%)	
N3	0 (0.0%)	7 (1.8%)		7 (1.6%)	0 (0.0%)	0 (0.0%)	
Unknown	0 (0.0%)	1 (0.3%)		1 (0.2%)	0 (0.0%)	0 (0.0%)	
P16/HPV status			0.014				0.481
Negative	22 (36.7%)	77 (20.0%)		99 (22.3%)	22 (36.7%)	16 (26.7%)	
Positive	11 (18.3%)	79 (20.5%)		90 (20.2%)	11 (18.3%)	14 (23.3%)	
Unknown	27 (45.0%)	229 (59.5%)		256 (57.5%)	27 (45.0%)	30 (50.0%)	

ECOG, Eastern Cooperative Oncology Group; PS, performance status; HPV, human papillomavirus.

**Table 2 cancers-15-00471-t002:** Treatment strategies by sex.

Treatment Strategy		Overall Cohort				Total N = 445	Propensity-Score-Matched Cohort				Total N = 120
		Female, n = 60		Male, n = 385			Female, n = 60		Male, n = 60		
Concurrent chemoradiotherapy	Induction CTx	25 (41.7%)	15	204 (53.0%)	88	229 (51.5%)	25 (41.7%)	15	27 (45.0%)	13	52 (43.3%)
	No induction		10		116			10		14	
Surgery	Induction CTx	30 (50.0%)	4	157 (40.8%)	28	187 (42.0%)	30 (50.0%)	4	31 (51.7%)	6	61 (50.8%)
	No induction		26		129			26		25	
Incomplete treatment	Induction CTx	5 (8.3%)	1	24 (6.2%)	19	29 (6.5%)	5 (8.3%)	1	2 (3.3%)	2	7 (5.8%)
	No treatment		4		5			4		0	
		*p* = 0.260					*p* = 0.585				

CTx, chemotherapy.

**Table 3 cancers-15-00471-t003:** Characteristics of treatment modalities in patients with locally advanced head and neck squamous cell carcinoma by sex.

Treatment		Sex			Number (%)
		Female	Male	*p*	
Induction chemotherapy		23 (14.6%)	135 (85.4%)		n = 158
Regimen	Docetaxel + cisplatin	13 (56.5%)	64 (47.4%)	0.529	77 (48.7%)
	Docetaxel + cisplatin + fluorouracil	6 (26.1%)	36 (26.7%)		42 (26.6%)
	Fluorouracil + cisplatin	4 (17.4%)	24 (17.8%)		28 (17.7%)
	Other	0 (0.0%)	11 (8.2%)		11 (7.0%)
Number of cycles	Median cycle [range]	3 [1–4]	3 [1–5]	0.906	3 [1–5]
Best overall response	Complete response	3 (13.0%)	22 (16.3%)	0.117	25 (15.8%)
	Partial response	12 (52.2%)	75 (55.6%)		87 (55.1%)
	Stable disease	8 (34.8%)	23 (17.0%)		31 (19.6%)
	Progressive disease	0 (0%)	15 (11.1%)		15 (9.5%)
Concurrent chemoradiotherapy		35 (58.3%)	270 (70.1%)		n = 305
Regimen	Weekly cisplatin	23 (65.7%)	152 (56.3%)	0.294	175 (57.4%)
	3-times-weekly cisplatin	6 (17.1%)	85 (31.5%)		91 (29.8%)
	Fluorouracil + cisplatin	3 (8.6%)	21 (7.8%)		24 (7.9%)
	Others	3 (8.6%)	12 (4.4%)		15 (4.9%)
Total radiation dose, Gy	Mean [95% CI], Gy	67 [44–70]	67 [43–72]	0.767	67 [43–72]
Best overall response	Complete response	19 (54.3%)	147 (54.9%)	0.861	166 (54.8%)
	Partial response	6 (17.1%)	59 (22.0%)		65 (21.5%)
	Stable disease	7 (20.0%)	42 (15.7%)		49 (16.2%)
	Progressive disease	3 (8.6%)	20 (7.5%)		23 (7.6%)

**Table 4 cancers-15-00471-t004:** Univariate and multivariate analyses for risk factors of overall survival in 60 female patients.

Characteristics	Univariate		Multivariate	
	Hazard Ratio (95% CI)	*p*	Hazard Ratio (95% CI)	*p*
Age	0.98 (0.94–1.02)	0.238		
ECOG PS				
2–3 vs. 0–1	Not calculable			
Smoking history				
Current or former vs. never	1.61 (0.35–7.36)	0.539		
Alcohol history				
Drinker vs. non-drinker	3.01 (0.78–12.10)	0.109	4.79 (1.15–19.92)	0.031
HPV status				
Positive vs. negative	0.38 (0.05–3.12)	0.370		
Primary tumor location				
Oropharynx	1 (reference)			
Oral cavity	1.74 (0.51–5.95)	0.380		
Hypopharynx	0.94 (0.11–8.44)	0.957		
Larynx	Incalculable			
Others	3.95 (0.88–17.77)	0.074	5.43 (1.40–21.08)	0.015
T classification	1.33 (0.81–2.17)	0.257		
N classification	1.75 (0.78–3.95)	0.176	1.95 (0.89–4.27)	0.093
Induction chemotherapy				
Yes vs. No	1.33 (0.48–3.67)	0.584		
Concurrent chemoradiotherapy				
Yes vs. No	0.93 (0.34–2.58)	0.892		
Surgery				
Yes vs. No	0.75 (0.27–2.07)	0.576		

HPV: human papilloma virus; PS: performance status.

## Data Availability

Data will be made available from the corresponding author on reasonable request.

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
