# Peer review of "Comparison of Treatment Patterns and Clinical Outcomes by Gender in Locally Advanced Head and Neck Squamous Cell Carcinoma (KCSG HN13-01)"

_cancers, 2023, doi:10.3390/cancers15020471_

Round 1
Reviewer 1 Report
It has been a pleasure to review the manuscript entitled "Comparison of treatment patterns and clinical outcomes by gender in locally advanced head and neck squamous cell carcinoma (KCSG HN13-01)". The paper is well written, methodology is accurate and results are sounded. Authors have outlined a sounded and appropriate discussion in length and contents. I would recommend its publication
Author Response
The paper is well written, methodology is accurate and results are sounded. Authors have outlined a sounded and appropriate discussion in length and contents. I would recommend its publication.
=> We appreciate your efforts to take the time for review.
Reviewer 2 Report
This is an interesting study about comparison of treatment patterns and clinical outcomes by gender in locally advanced head and neck squamous cell carcinoma. The study included 445 patients. To minimize the observed confounding, the propensity score matching was utilized.
The paper is well written. However, some issues remain.
Please add some more data about sex disparities in published studies on head and neck cancer in the Introduction section.
Disease Free Survival and Disease Specific Survival can be helpful to better understand sex similarities or differences. Please add them.
The authors must specify that recurrences were excluded.
Unknown smoking and alcohol history percentages are high. It is possible to reduce them?
I think that a better matching for tumor sites between men and women should be achieved.
How many patients who underwent surgery had adjuvant radiotherapy or chemoradiotherapy?
Reviewer 3 Report
This paper discusses the gender role in HNSCC treatments in Republic of Korea. I thank the editor to permit me to read and comment this article.
I have some major comments that I analysed directly on the paper (see pdf in attachment). In particular I think that there are some great bias (HPV status not in all pts, pts selection, few patients analysed).

Round 2
Reviewer 2 Report
Thanks for improving the manuscript.
Reviewer 3 Report
I thank the authors for the opportunity to read this revised version.